# TIDMAD: TIME SERIES DATASET FOR DISCOVERING DARK MATTER WITH AI DENOISING

## ABSTRACT

Dark matter makes up approximately 85% of total matter in our universe, yet it has never been directly observed in any laboratory on Earth. The origin of dark matter is one of the most important questions in contemporary physics, and a convincing detection of dark matter would be a Nobel-Prize-level breakthrough in fundamental science. The ABRACADABRA experiment was specifically designed to search for dark matter. Although it has not yet made a discovery, ABRACADABRA has produced several dark matter search results widely endorsed by the physics community. The experiment generates ultra-long time-series data at a rate of 10 million samples per second, where the dark matter signal would manifest itself as a sinusoidal oscillation mode within the ultra-long time series. In this paper, we present the TIDMAD — a comprehensive data release from the ABRACADABRA experiment including three key components: an ultra-long time series dataset divided into training, validation, and science subsets; a carefully-designed denoising score for direct model benchmarking; and a complete analysis framework which produces a community-standard dark matter search result suitable for publication as a physics paper. This data release enables core AI algorithms to extract the signal and produce real physics results thereby advancing fundamental science. The data downloading and associated analysis scripts are available at https://anonymous.4open.science/r/TIDMAD.

## 1 INTRODUCTION

The quest to uncover the nature of dark matter is one of the biggest challenges in contemporary physics. Several key observations in astrophysics and cosmology have confirmed the existence of dark matter, which constitutes approximately 85% of all mass in the universe (Rubin & Ford, 1970; Tyson et al., 1998; Tegmark & et. al., 2004; Aghanim & et. al., 2020; Adams & et. al., 2023). However, dark matter has never been detected by any detector on Earth. Because the composition of dark matter is unknown, theoretical physicists propose various dark matter candidates — hypothetical particles that can be characterized by their physical parameters. Experimental physicists then design experiments to search for these candidates. A convincing detection of any dark matter candidate would be a Nobel-Prize-level breakthrough in fundamental science, but even if nothing is detected, the null results still play a significant role in advancing our understanding of physics by setting limits within the physical parameter space. This means that a particular experiment has eliminated the existence of a dark matter candidates within these limits and does not have sufficient sensitivity to test outside these limits. These limits can be reciprocally used by theoretical physicists to propose better dark matter candidates, thereby improving our understanding of this mysterious constituent of our universe.

Attributable to its extremely rare interactions with normal matter, the signal of dark matter is often submerged in a sea of noise from various sources internal and external to the experimental apparatus. While the signal shape is extremely well parameterized, modeling backgrounds with traditional methods is intractable given they are composed of both transient and persistent noise sources, frequency dependant and independent sources, phase coherent and incoherent noise, and amplifier dependant and independent noise. Machine learning (ML) offers a promising means to reduce this noise. Advancements in denoising techniques using ML algorithms have the potential to significantly improve dark matter analyses Saleem et al. (2023). These techniques enable the detection of weaker dark matter signals, or in the case of no observation, the setting of stronger limits. In

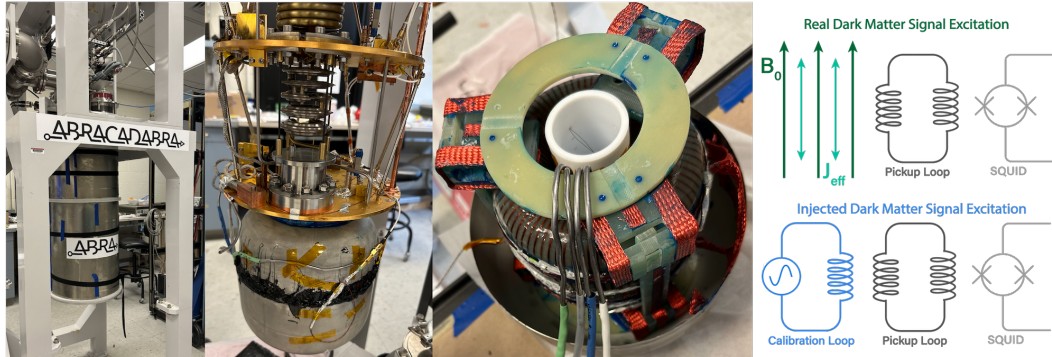

Figure 1: Left to right: ABRA dilution fridge with outer vacuum cans on; Coldest stage of ABRA fridge above shielded 1T superconducting torroidal magnet; Interior of ABRA magnet including pickup and calibration loop wires existing the center of the magnet; Effective circuit diagram for both dark matter and injected signals.

other words, improvements in data denoising directly enhance the scientific reach of dark matter experiments. In this paper, we present an ultra-long time series dataset produced by a real dark matter detector: ABRACADABRA (A Broadband/Resonant Approach to Cosmic Axion Detection with an Amplifying B-field Ring Apparatus, abbrev. ABRA). ABRA is the world leading sub-$\mu eV$ dark matter experiment that pioneered the quantum enabled lumped element dark matter detection technique Ouellet & et. al. (2019a;b); Salemi & et. al. (2021). We operated the ABRA detector in February 2024 to obtain a special time series dataset for these studies: TIDMAD (TIme series dataset for discovering Dark Matter with Ai Denoising). It can be partitioned into three parts: (1) training data, (2) validation data, and (3) science data.

The training data include time series data where a dark matter-like signal is injected by hardware. If dark matter enters ABRA, it will manifest itself as a sinusoidal oscillation mode within the time series; therefore, the injected signals are also sinusoidal oscillations at specific frequencies. Both the detected (noisy) time series and the injected (ground truth, clean) time series are provided with one-to-one temporal correspondence. This allows the training of machine learning algorithms to denoise the detector data and recover the injected signal. The validation data is used to produce Benchmark 1: Denoising Score, see Section 4.1. Algorithms that effectively dampen the detector noise while amplifying the injected signal will achieve a better denoising score. The science data is collected without the injected signal with an extended duration to produce Benchmark 2: Dark Matter Limits, as discussed in the previous paragraph. The limit generation procedure is detailed in Section 4.2. The scientific data are titled to reflect their use in producing real, community-standard physics results that are suitable for presentation in scientific journals. Several traditional and deep learning denoising algorithms are presented in Section 3 and Appendix C, where the resulting denoised data is benchmarked against the raw, un-denoised detector data.

## 1.1 AXION DARK MATTER AND ABRACADABRA

In recent years, the axion has emerged as one of the leading dark matter candidates as a result of its theoretical elegance. Axions interact with normal matter via electromagnetism, which can be characterized by a physics parameter $g_{a\gamma\gamma}$. Arising from its small mass $m_a < 1eV$ ($10^{-6}$ times smaller than electron), axions acts as a classical field oscillating at a frequency $f_a = m_a/2\pi$. Astrophysical measurements determine that the Earth exists in a bath of dark matter with a known local density of $\rho_{DM}$ de Salas & Widmark (2021).

The latest advancements in quantum detector technology have facilitated new avenues to search for the axion. ABRA is one of the novel detectors designed to search for axions leveraging these advancements in quantum technologies Ouellet & et. al. (2019b). ABRA capitalizes on the fact that we are immersed in a bath of a feebly electromagnetically-interacting, oscillating dark matter field to detect this elusive particle. Specifically, in the presences of a static magnetic field $\mathbf{B}_0$, the axion, henceforth referred to as dark matter, induces an oscillating magnetic field $\mathbf{B}_a$. Thus, to detect dark

matter ABRA provides a strong magnetic field $B_0 = 1T$ and uses a superconducting pickup loop to read out the oscillating dark matter signal. Read out by a superconducting quantum interference device (SQUID), the pickup loop detects the dark matter signal as a time-oscillating current given by

$$\mathbf{J}_{eff} = g_{a\gamma\gamma}\sqrt{2\rho_{DM}}\mathbf{B}_0 cos(m_a t) \tag{1}$$

where the two parameters that define the theory, the coupling $g_{a\gamma\gamma}$ and mass $m_a$, appear as the relative strength of the signal and oscillation frequency respectively Ouellet & et. al. (2019a). The total signal power expected in our detector is given by

$$A \equiv \langle|\Phi_a|^2\rangle = g_{a\gamma\gamma}^2\rho_{DM}\mathcal{G}^2V^2B_{max}^2 \tag{2}$$

where $\mathcal{G} \approx 0.0217$ is a geometric coupling, $V \approx 890\text{cm}^3$ is the magnetic field volume, and $B_{max} \approx 1$ T is the maximum static magnetic field Ouellet & et. al. (2019a).

## 1.2 TIDMAD CONSTRUCTION

In the classical analysis, we use a calibration procedure to determine the end-to-end response of our system for different signal frequencies. As shown in Figure 1, the ABRA detector contains a toroidal magnet equipped with both a pickup loop and a calibration loop. During calibration, we first inject a fake dark matter signal into the calibration loop at a specific frequency. This generates a sine wave with a known amplitude and frequency, creating a dark matter-like flux with our pickup loop. Finally, this flux is detected by the SQUID sensor for detector calibration.

The dark matter signal injected into the calibration loop by the signal generator follows the form prescribed by axion theory, as shown in Equation 1 and derived in Appendix A. We specifically choose to inject sine waves with frequencies from 1.1 kHz to 4.9 MHz, corresponding to axion masses $m_a = [0.005, 17]$ neV, to target the mass range that our experimental hardware is designed to detect. The injected signals were all set to an amplitude of 50 mV to ensure a reasonable signal-to-noise ratio. A total of 309 different frequencies were sequentially stepped through, from 1.1 kHz to 4.9 MHz, simulating 309 distinct axion masses in our detector hardware.

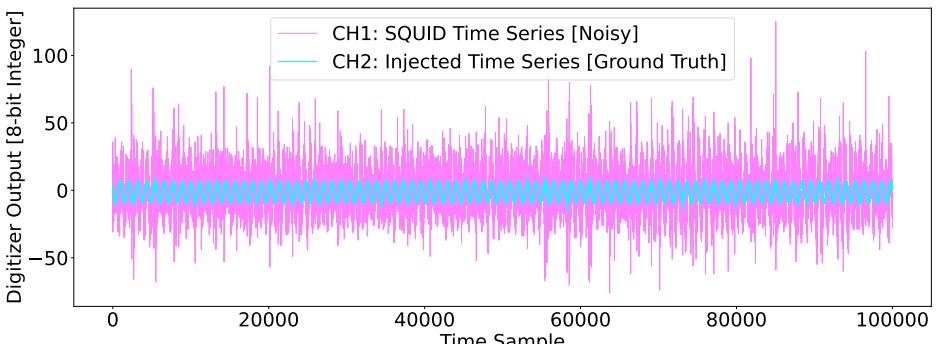

Figure 2: 10-millisecond snapshot of the time series in TIDMAD training dataset.

The TIDMAD dataset presented in this work is inspired by this calibration procedure. The ABRA detector hardware enables us to simultaneously record two types of ultra-long time series: the one injected into the calibration loop (referred to as the "injected time series") and the one detected by the SQUID sensor coupled to the pickup loop (referred to as the "SQUID time series"). As shown in Figure 2, the injected time series exhibit a clear sinusoidal oscillatory signal, which can be considered the ground truth. Meanwhile, the SQUID time series contains the same ground truth submerged within a sea of detector noises. The two time series are exactly aligned at every time step. This defines the signal recovery task: a model could be applied to the SQUID time series to reproduce the injected signal in the injected time series. A model trained on this task will be efficient in rejecting noise of different kinds while retaining the dark-matter-like signal within the detector. We then collected a science dataset where no fake dark matter signal is injected. The trained denoising model can then be applied to the SQUID time series of the science dataset. If a

sinusodial signal is found after denoising, it could potentially be a real dark matter particle entering the detector.

## 2 DATASET DESCRIPTION

The data presented in this paper were acquired using the ABRACADABRA detector. The overall schematics of the data is shown in Figure 3. All data are saved as a series at 10 MS/s (Megasample per second), where each sample is a 8-bit integer ranging from -128 to 127. These integers can be converted into a physics units of mV (millivolts) with a scaling factor of $40/128$. The procured datasets are stored at Open Science Data Federation (OSDF) Weitzel et al. (2017) in .hdf5 format and can be accessed via the download_data.py script in the github repository provided.

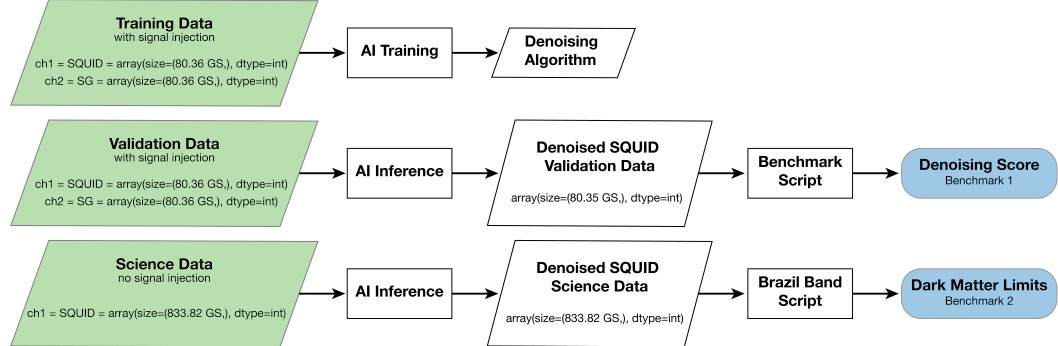

Figure 3: TIDMAD data flow explaining how data splits (green) result in benchmarks (blue). Rectangles correspond to provided scripts. Left to right, top to bottom the scripts are train.py, inference.py, benchmark.py, process_science_data.py, brazilband.py

**Training dataset:** The training dataset contains 80.23 Gigasamples of time series data, corresponding to roughly two hours of data collection. For training purposes, we injected a fake dark matter signal into the hardware as discussed in Section 1.2. The injected signal scans through dark matter frequencies from 1100 Hz to 5 MHz at two different amplitudes: 50 mV (standard) and 10 mV (weak). Only the standard injection was considered in the rest of this paper. However, the weak injection data is also available to download by using the additional $-w$ flag with the download_data.py script, providing a more challenging scenario for signal recovery. All training data are partitioned into 20 files. Each file contains two channels: the injected time series is saved in the second channel (CH2), while the SQUID time series is saved in the first channel (CH1). As discussed in Section 1.2, the training task is to recover the CH2 ground truth time series using the CH1 noisy time series as input.

**Validation dataset:** The validation dataset, consisting of 80.23 Gigasamples, has the same format as the training dataset. The only difference is that the validation dataset was independently collected at a different time using the same detector apparatus, making it an out-of-sample dataset with slightly altered noise conditions. After training, users can perform model inference by running the inference.py script to denoise the CH1 time series while preserving the injected signal. The denoised SQUID time series in CH1 and the injected time series in CH2 are then processed through the benchmark.py script. To determine the efficacy of the model's denoising, a benchmarking score called 'denoising score' is calculated, which will be discussed in detail in Section 4.1.

**Science dataset:** The science dataset comprises 833.82 Gigasamples of time series data collected over a 24-hour period. This data is distributed across 208 .hdf5 files. Unlike the training and validation datasets, there are no injected signals, meaning that only the CH1 time series is saved per file. The inference task for ML models is to denoise the SQUID time series in CH1. The denoised science data is then analyzed to obtain a dark matter limit, which will be discussed in Section 4.2.

## 3 EXPERIMENTS

We experimented five different denoising algorithms including two traditional algorithms and three deep learning models. The traditional algorithm can be directly applied to the validation dataset using `inference.py`, while deep learning models need to be trained first on the training dataset with `train.py`. The five algorithms are listed below:

- **Moving average**: a simple moving average with a window size of 100, implemented using the numpy convolve function.

- **Savitzky-Golay filter**: with a window size of 100 and a polynomial order of 11.

- **FC net**: an autoencoder architecture designed for transforming input data. This model consists of an encoder and a decoder. The encoder encodes the input data into a low-dimensional representation, while the decoder reconstructs the original data from this encoded representation. Both the encoder and decoder are composed of multiple fully-connected layers and activation layers. FC-Net outputs a single float point number at each time step, and the training is conducted by minimizing the mean square error between this float point number and corresponding ground truth time series at every sample.

- **PU net**: a deep learning architecture based on the UNet architecture Ronneberger et al. (2015). U-Net uses convolution layers as encoder and deconvolution layers as decoder, with contracting paths established between each pair of convolutional and deconvolution layers at the same level. This allows information at different encoding levels to flow to the decoding part. Positional encoding layers are introduced at all encoder layers to enhance the model's ability to understand positions in the time series. Since every sample of the ground truth time series has to be 8-bit integers ranging from -128 to 127, we require the model to output a 256-class classification decision at every time step, where each class corresponds to one possible output value. This effectively redefines the denoising task into a semantic segmentation task.

- **Transformer**: the transformer utilizes a self-attention mechanism to capture long-distance dependencies in sequences Vaswani et al. (2017). After processing by the multi-layer Transformer encoder, the model effectively extracts features and represents the input sequence. Finally, the encoded sequence is mapped to the output dimension through a linear layer for the same 256-class classification decision as PU-Net. Positional encoding is also added before the time series is fed into Transformer layers. Both PU-Net and Transformer are trained using Focal Loss to handle class-imbalanced segmentation labels Lin et al. (2017).

The benchmarking results of these models are discussed in Section 4. There are two additional constraints for the deep learning models. First, because of memory constraints, we segment the ultra-long SQUID and injected time series into smaller segments before feeding them into each model. The exact segment sizes are outlined in Table 1. Secondly, as a result of the large range of injected frequencies, training multiple versions of the same model to handle different frequency ranges is necessary. Both limitations and additional details of the deep learning models are discussed in Appendix B.

## 4 EVALUATION METRICS

We developed two benchmarking criteria to evaluate the performance of denoising algorithms. Benchmark 1: Denoising Score provides a quantitative measure of denoising performance based on the signal-to-noise ratio. This score is designed to be linear with respect to the noise level and equal to one when no denoising is applied. While Benchmark 1 offers a quick, straightforward assessment of model performance, it lacks direct relevance to fundamental science. To bridge this gap, we developed Benchmark 2: Dark Matter Limit, which directly links AI algorithms to community-standard physics result by automating the entire dark matter analysis on the science dataset. This benchmark allows AI algorithms to directly improve the physics reach of dark matter detectors.

## 4.1 BENCHMARK 1: DENOISING SCORE

The denoising score is a modified signal-to-noise ratio (SNR) of the denoised CH1 SQUID time series. It is calculated over the validation dataset by first segmenting both the injected and SQUID time series into one-second segments. Each second of the time series is transformed into a power spectral density (PSD) using a squared fast Fourier transform. This frequency domain data records signal power at each frequency – $PSD(\nu)$. Since the injected dark matter signal is a clean sinusodial oscillation, it should appear as a single-bin peak ($\nu_0$) in the PSD, while noise in SQUID time series is distributed across all frequency bins. The location of $\nu_0$ is identified by the PSD of the injected time series (ground truth) as the largest single bin peak relative to its nearest neighbors.

$$\nu_0 = \underset{\nu}{\mathrm{argmax}}\Big(PSD_{\mathrm{Injected}}(\nu) - (PSD_{\mathrm{Injected}}(\nu - df) + PSD_{\mathrm{Injected}}(\nu + df))\Big) \quad (3)$$

where $df$ is the sampling frequency of $10^{-7}$ Hz. Once $\nu_0$ is identified in the injected time series, the the signal region is defined by selecting $n_{sig} = \pm 1$ bins around the signal frequency to account for spectral leakage. Similarly, the noise region is defined by selecting $n_{bkgd} = \pm 50$ bins outside of the signal region. By taking the ratio of the PSD in the signal region to that in the noise region, we acquire the SNR for each one-second segment PSD.

$$SNR_i = \left(\frac{P_{sig}}{P_{noise}}\right)_i = \sum_{\nu=\nu_0-\nu_{sig}}^{\nu_0+\nu_{sig}} PSD_i(\nu) \Bigg/ \sum_{\nu=\nu_0-\nu_{bkg}}^{\nu_0+\nu_{bkg}} PSD_i(\nu) \quad (4)$$

Multiplying by the sampling frequency ($df$) turns bin range ($n_{sig,bkg}$) to frequency range ($\nu_{sig,bkg}$).

The hardware setup includes a bandpass filter between the pickups and the digitizer, resulting in a frequency dependence for the signal magnitude in both the injected and the SQUID time series. To account for this, we first calculate the normalized injected SNR:

$$(SNR'_{\mathrm{Injected}})_i = \frac{(SNR_{\mathrm{Injected}})_i}{\max(SNR_{\mathrm{Injected}})} \quad (5)$$

The SQUID SNR then gets multiplied to the corresponding, normalized SQUID SNR in the the same one-second segments, and then summed over all one-second segments to produce $\Lambda$ defined below:

$$\Lambda = \left(\frac{1}{n}\sum_{i=0}^{n}(SNR_{\mathrm{SQUID}})_i \times (SNR'_{\mathrm{Injected}})_i\right) \quad (6)$$

We examine the validity of $\Lambda$ to represent denoising efficiency through a study involving added Gaussian noise. In the ABRA detector, there are a multitude of independent, random noise sources internal and external to the detector thus the noise in our data follows a Gaussin distribution. In this study, Gaussian noise is imposed on the injected time series, and $\Lambda$ are calculated at different levels of Gaussian noise amplitudes and standard deviations. As shown in Figure 4 (right), $\Lambda$ exhibits an exponential decay trend with increased Gaussian noise amplitude. To establish a linear correlation between the denoising score and noise, we apply a logarithmic transformation to $\Lambda$ to calculate the final denoising score:

$$\text{Denoising Score} = log_{5.27}\Lambda \quad (7)$$

The base of the logarithm is chosen to be 5.27 so that the denoising score equals 1 for the raw SQUID time series over the validation dataset (i.e., when no denoising algorithm is applied). We further examined this denoising score over a range of imposed Gaussian noise amplitude and STD, and observed a smooth linear response as shown in Figure 4 (left).

This denoising score is implemented in the provided script, benchmark.py, which takes as input the denoised SQUID time series of the validation dataset produced by model inference. The script is designed for parallelization and takes about 30 minutes to run on an 8-core CPU node. To further reduce the time required for calculating the denoising score, we defined this second-by-second scan as the Fine Score and introduced a new Coarse Score Denoising Score. The Coarse Score is a tenfold downsample of the full Fine Scan thereby providing a fast benchmarking score that users can leverage to get a rough estimate of model performance in 10% the computational time.

Table 1 shows the denoising scores for all algorithms discussed in Section 3. The case with no denoising is shown in the first row, with its fine denoising score calibrated to 1. We observed that

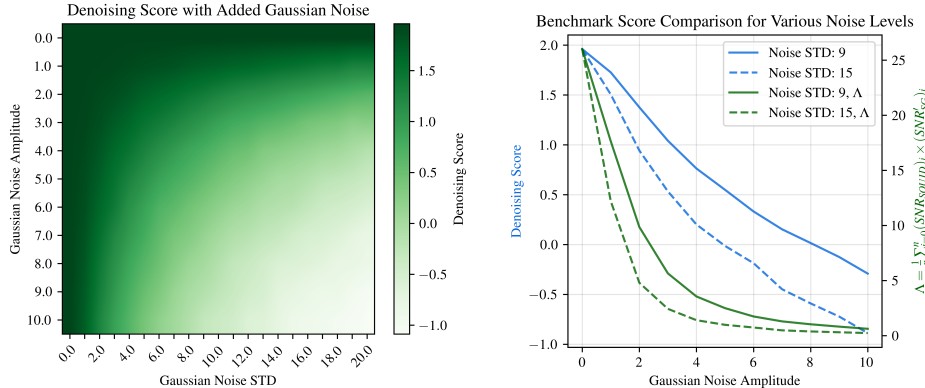

Figure 4: Left: The color bar represents the denoising score for 20s of raw data with added Gaussian noise showing that noisier data results in a lower score. Right: Denoising score and $\Lambda$ for 20s of raw data with added Gaussian noise of variable noise amplitude. The exponential behavior of $\Lambda$ can clearly be seen in contrast to the smooth linearity of the denoising score.

Table 1: Fine and coarse denoising score for raw data, traditional algorithms, and trained ML models

| Algorithms | Segment Size | Parameters | Fine Score | Coarse Score |
|---|---|---|---|---|
| None | | | 1.00 | 1.10 |
| Moving Average | $1 \times 10^6$ | window = 100 | 0.52 | 0.64 |
| SG Filter | $1 \times 10^6$ | window = 100, order = 11 | -2.77 | -2.35 |
| FC Net | $4 \times 10^4$ | See Appendix B | 6.43 | 6.55 |
| PU Net | $4 \times 10^4$ | See Appendix B | 3.69 | 3.84 |
| Transformer | $2 \times 10^4$ | See Appendix B | 3.95 | 4.18 |

in all cases, the coarse denoising score is slightly higher than the fine denoising score. Based on the results, all traditional algorithms decrease the denoising score as time domain averaging erases high-frequency signals in the region of interest. Meanwhile, all deep learning algorithms efficiently boost the denoising score. Surprisingly, we observed that the FC Net model achieved the best performance with a denoising score of 6.43.

## 4.2 BENCHMARK 2: DARK MATTER LIMIT

The second benchmark empowers algorithm creators with the capability to conduct a community-standard dark matter search using the science dataset. Following the discussion in the Introduction, the two physics parameters for the dark matter candidate in this paper are the dark matter mass ($m_a$) and the dark matter to electromagnetic coupling ($g_{a\gamma\gamma}$). Null results from different dark matter experiments place limits within this parameter space expressed by the shaded regions in Figure 5. In the physics community, a better dark matter limit is represented by pushing towards lower values of $g_{a\gamma\gamma}$ at different $m_a$. We provide a comprehensive tool necessary for performing the statistical analysis to produce dark matter limits in Figure 5. The dark matter mass $m_a$ is directly proportional to frequency, therefore the limit-setting procedure is repeated for 11.1 million independent $m_a$ from 0.4 neV (100 kHz) to 8 neV (2 MHz). The dark matter limit at each $m_a$ is obtained using a frequentist log-likelihood ratio test statistic (TS), with the results depicted as regions in Figure 5. The detail of this analysis can be found in Appendix C.

This analysis is performed twice on the SQUID time series of the science dataset: once without any denoising algorithm which produces the ABRA-TIDMAD Raw limit, and once with FC Net, the top-performing denoising algorithm, which produces the ABRA-TIDMAD Denoised limit. These limits can be directly compared to the previous world-leading ABRA Run 3 limit, limits obtained by

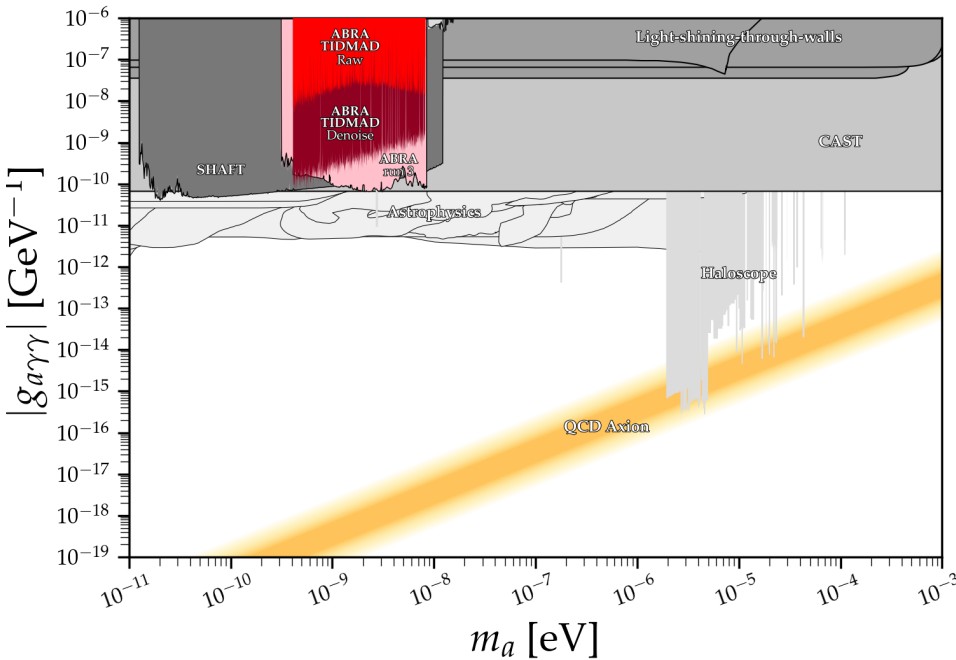

Figure 5: Plotted alongside the present, state-of-the-art axion dark matter limits (grey) are the $2\sigma$ exclusion limits for ABRA Run 3 (pink) Salemi & et. al. (2021), ABRA-TIDMAD Raw limit (red), and ABRA-TIDMAD Denoised limit from the trained FC net (maroon). Shaded regions correspond to pairing of dark matter model parameters ($m_a$, $g_{a\gamma\gamma}$) that are ruled out by the specified experiment or observation, and the bright yellow region indicates theoretical predictions (See Appendix D for details). The plotting script is modified from the public `AxionLimits` repository O'Hare.

other dark matter experiments, as well as theoretical predictions Salemi & et. al. (2021). While the ABRA-TIDMAD Denoised limit does not outperform the ABRA Run 3 limit because of hardware and time constraints, it is evident that denoising algorithms significantly improved the dark matter limit by 1-2 orders of magnitude across different $m_a$. Although the size of the ABRA-TIDMAD science dataset is only 1% of the ABRA Run 3 science dataset, the AI denoising algorithm boosted the ABRA-TIDMAD limit to nearly the same level as ABRA Run 3 and even surpassed the ABRA Run 3 limits at small $m_a$.

## 5 LIMITATIONS AND APPLICATIONS

**Hardware and datataking period:** As illustrated in Figure 5, the baseline models fail to surpass the results of ABRA Run 3. This is attributed to hardware limitations and changes since the last data run, including (1) replacing the dark matter pickup cylinder with a pickup loop, consequently reducing the geometric coupling to the dark matter signal, and (2) reducing the data taking period to 24 hours from three months. The decision to implement these changes was driven by the aim to enable ABRA to simultaneously search for dark matter and gravitational waves, thereby enhancing the scientific scope of the experiment. The shortened data taking duration was necessitated by operational constraints of the dilution refrigerator. Because the signal-to-noise ratio scales as the fourth root of integration time, we can increase ABRA's sensitivity to dark matter by increasing the data taking period Ouellet & Bogorad (2019). Another more efficacious way to increase this signal-to-noise ratio is improving our denoising with ML; a doubling of our noise reduction represents a *16x* speed up our data taking time revealing the out-sized return on investment in denoising techniques when compared to increased detector run time.

**Null result vs. potential discovery:** In Section 4.2, we discussed how to set a dark matter limit using the provided analysis scripts. This script assumes a null result as no $5\sigma$ dark matter candidates were identified in this region of parameter space by ABRA Run 3. Therefore, we assume a null

result for this much shorter (24 hr) data taking. This assumption enables us to establish upper limits on the coupling parameter $g_{a\gamma\gamma}$ for every mass point. As shown in Figure 5, the ABRA-TIDMAD Raw limit without denoising covers a smaller region than ABRA Run 3 Salemi & et. al. (2021).

However, with the denoising algorithm applied to the 24-hour science data, ABRA-TIDMAD could potentially reach beyond the ABRA Run 3 region, where a discovery of dark matter is possible. In this paper, we focused on increasing the experimental sensitivity and setting exclusion limits. A straightforward modification to the interpretation of the TS would unlock the discovery potential of this analysis framework. Future efforts will focus on employing a more extensive dataset and implementing dark matter discovery analysis code. Given that discovering dark matter would be a Nobel Prize-level breakthrough, it is crucial to not only claim discovery but also to convince the scientific community of its validity. If TIDMAD users find any anomalous signals in the science dataset, please contact the authors for further investigation and understandings of systematic uncertainties.

**Generalizability:** Axion dark matter is a specific subset of wave-like dark matter candidates, making the techniques developed in this paper broadly applicable to a wide range of wave-like dark matter experiments. Other axion dark matter experiments, including but not limited to ADMX Braine et al. (2020), HAYSTAC Backes et al. (2021), and CASPEr Budker et al. (2014), also produces long time series data and search for similar peaks in frequency domain; any AI algorithm developed upon TIDMAD can be easily adapted and applied to these experiments. Furthermore, time series denoising algorithms are crucial for extracting wave-like signals in various areas of physics. In astrophysics, gravitational wave searches involve detecting chirp signals with durations on the order of seconds Abbott et al. (2016), often buried within detector noise. In nuclear physics, denoising can enhance the efficiency of HPGe detectors Anderson et al. (2022) and bolometer detectors Vetter et al. (2024). Advancements in denoising can be deployed across a suite of these physics experiments.

While the data released in this paper was tailored to our specific problem statement and benchmarks in physics, these ultra-long time series datasets have the potential to benefit a wide range of applications beyond physics. Similar to TIDMAD, many other scientific domains involves time series datasets exhibiting relatively uniform frequency characteristics, with the primary analytical task focused on extracting signals from these time series. Examples include pulsar timing from radio observatory data (astronomy)Hobbs et al. (2006), detecting seismic arrivals above background noise (geology)Webb (2002), identifying sea surface and near-surface temperature anomalies (climate science)Smith et al. (2008), and recognizing atrial fibrillation among noises and other rhythms in short-term ECG recordings (health science)Clifford et al. (2017). If a foundation model were to be developed for general time series analysis in science, our frequency-rich, detector-generated, long time series data could provide a uniquely abundant source of spectral complexity.

## 6 CONCLUSIONS AND OTHER WORKS

We present TIDMAD, the first dataset and benchmark designed to yield a community-standard dark matter search result. TIDMAD includes all necessary inputs and processing to train time series denoising algorithms and produce a science-level dark matter limit. Through a series of experiments, we developed three ultra-long time series deep learning algorithms, benchmarked their ability to recover hardware-injected signals, and set dark matter limits. Clear performance improvements were demonstrated on both benchmarks. Our future work will focus on enhancing the denoising algorithm to achieve better dark matter limits, expanding to other nuclear and particle experiments, and embedding these algorithms onto FPGA chips for real-time denoising during data taking.

The aim of this data release is to enable the ML community to use TIDMAD to develop algorithms tailored for data with highly coherent embedded signals. This development would not only extend the experimental reach of dark matter searches, leading to improved dark matter limits, but also allow the AI/ML community to make direct scientific advancements. This transparency aims to foster greater collaboration between the ML and particle physics communities, benefiting both fields.

## ETHICS STATEMENT

This research does not involve any human subjects, and all data utilized in this study were collected exclusively through physics hardware. The experimental data were obtained following rigorous scientific protocols to ensure accuracy and reliability. We have adhered to ethical standards in data handling, analysis, and reporting, ensuring the integrity and reproducibility of our findings. No ethical concerns are associated with the use of the hardware or data collection methods employed in this research.

## REPRODUCABILITY

The information necessary to reproduce all of the results in this paper are thoroughly catalogued throughout the paper. Reproducing our results can be broken into seven parts (1) collecting data (2) accessing data (3) training denoising algorithms (4) deriving denoising benchmark (5) evaluating algorithms on denoising benchmark (6) deriving dark matter limit benchmark (7) evaluating algorithms on dark matter limit benchmark.

To reproduce the ABRACADABRA hardware configurations used in collecting this data, see Appendix E.3. The data composition, storage, maintenance, and distribution information is presented in Appendix E with data access described in Section F. To reproduce our trained denoising algorithms, see Section 3. In our code repository is the script `train.py` which will reproduce the trained models. To derive the denoising benchmark, please see Section 4.1. The implementation and calculation of this benchmark is also in our code repository `benchmark.py`. To evaluate the algorithms with the denoising benchmark, the code is provided in our repository using scripts `inference.py` and `benchmark.py`. For a full derivation of our dark matter limits, please see Appendices A and C. To reproduce our dark matter limits, one can use our scripts `process_science_data.py` and `brazilband.py`. All of these scripts are in our public repository `https://anonymous.4open.science/r/TIDMAD`.

For detailed instructions on how to download and process our data to produce all results in this paper using our provided scripts, please see the `README.md` in our public repository.

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

## A DARK MATTER SIGNAL AND SIGNAL INJECTION

Axion dark matter appears in our detector as a time oscillating current with two free model parameters $(m_a, g_{a\gamma\gamma})$ described by Equation 1. This signal can be fully derived from axion theory and is extremely well-defined. While a full derivation of this equation is presented in Reference Ouellet & Bogorad (2019), schematically, this theoretical prediction for the dark matter signal in our detection comes from four phenomena – the omnipresent axion field, axion's interactions with electromagnetism, the specific geometry of our detector, and the velocity distribution of the dark matter.

Under certain conditions, axions are created in the early universe via the misalignment mechanism Dine & Fischler (1983); Preskill et al. (1983). When axions are created they produce a time oscillating omnipresent axion field with an oscillation frequency equal to $m_a$ and phase coherence. If axions are dark matter then the abundance of axions and the strength of this axion field are set by our astrophysical observations of dark matter density de Salas & Widmark (2021).

The axion field can interact with other physical forces including electromagnetism. If an axion field collides with an electromagnetic field, some of the axions will be transformed into photons. These photons produce a secondary electromagnetic field – an axion-induced electromagnetic field. In the geometry of our detector, the axion-induced electromagnetic field is read out with a pickup wire which samples the field like a typical radio antenna. This means the time oscillating axion field turns into a time oscillating current in our detector (i.e. our dark matter signal).

If earth were stationary, sitting in a uniform bath of dark matter, then the dark matter signal frequency would be exactly the axion field oscillation frequency (i.e. the axion mass) and completely coherent. However, the earth is moving within the Milky Way galaxy and thus the axion field gains a velocity with respect to earth at about $v_{DM} \sim 220km/s$. Doppler shifting spreads this frequency such that it is distribution around the original field frequency $\Delta f = v_{DM}^2 f$ Ouellet & Bogorad (2019). This frequency distribution is six orders of magnitude smaller than the signal frequency, therefore can be treated a coherent, single frequency sine-wave to good approximation. For signal injection, this single frequency approximation is used while for the dark matter analysis, we model the full frequency distribution.

Thus, we have established that the dark matter signal in the ABRA detector is approximately sine-wave current. The frequency of this oscillating signal is a model parameter meaning if we knew the axion mass, this frequency would be set. However, theoretical models point to a range of possible axion masses, not a singular value. Ideally we would create a dark matter detector that could search the entire range of valid dark matter masses, but experimental constraints such detector size, configuration, and readout electronics preclude this possibility. Instead, experiments must be tailored to search for smaller areas of the axion mass parameter space with ABRA the detector being specifically designed to target $m_a = 0.4 - 10$ neV Ouellet & et. al. (2019a).

The second free signal parameter is the sine wave amplitude which is proportional to $g_{a\gamma\gamma}$, the strength of the axion's interactions with electromagnetism. Theoretical calculations constrain this parameter to $g_{a\gamma\gamma} = Cm_a$ where $C = [-0.39, 0.22]$ depending on the theory Shifman et al. (1980); Dine et al. (1981). Theoretically motivated axion couplings can be seen in Figure 5 as the gold band in the $m_a, g_{a\gamma\gamma}$ parameter space. Experimentally, the goal is to detect ever smaller signal amplitudes to reach lower values of $g_{a\gamma\gamma}$.

To inject a fake signal into the hardware, we replicate the signal current, given by Equation 1, with a signal generator. The signal generator is connected to a calibration loop depicted in Figure 1 designed to mimic an axion field incident on the detector. We specifically choose to inject sine waves with frequencies from 1.1 kHz to 4.9 MHz, $m_a = [0.005, 17]$ neV, to contain the masses our experimental hardware was built to target. The injected fake signals we used all have amplitudes of 50 mV to achieve a reasonable signal-to-noise ratio. While injecting smaller fake signal amplitudes would effectively simulate dark matter candidates with smaller electromagnetic couplings, fake signals smaller than 50 mV are difficult to detect with traditional techniques. Though smaller couplings provide an interesting ML task, our denoising score benchmark is predicated on finding injected signals with traditional techniques and subsequently we did not use smaller fake signal amplitudes. However, we did take data injected with signals ranging from 1.1 kHz to 4.9 MHz at a signal am-

plitude of 10 mV. While this data was not used in our benchmark creation or model training, it is publicly available (see Datasheet) for an extra challenge.

To summarize, our signal injection scheme involves exciting hardware with a sine wave from a signal generator. We sequentially step through 309 different frequencies from 1.1 kHz to 4.9 MHz to simulated 309 different axion masses in our detector hardware all with an amplitude of 50 mV so that the fake signal is visible above the detector noise floor.

## B   DETAILS AND LIMITATIONS OF DEEP LEARNING MODEL

There are two special treatments we took to train the three deep learning models:

**Training segmentation:**    To feed the time series data into limited GPU memories, the training time series are segmented into 4 milliseconds. This imposes a fundamental lower limit on the frequencies that the models can detect. While this lower limit, approximately 250 Hz given the sampling frequency of 10 MS/s, is relatively small, it does establish a foundational lower threshold for frequency resolution in the model. The transformer model requires additional memory, therefore we have to further reduce the segment to 2 millisecond or 500 Hz. Both of these limits are well below the dark matter search range: 0.4 neV (100 kHz) to 8 neV (2 MHz).

**Frequency splitting:**    Since the injected dark matter signal spans two orders of magnitude in frequency, the observed features in the injected time series significantly vary. During training, we noticed that a single deep learning model to denoise the entire dataset would fail to generalize across the different injected frequency ranges. To address this issue, we trained four deep learning models per architecture, each focusing on a specific frequency range: the first covering the low-frequency regime (training/validation files 0-3), the second covering the mid-low regime (training/validation files 4-9), the third covering the mid-high regime (training/validation files 10-14), and the fourth covering the high-frequency regime (training/validation files 15-19). During the benchmark 1 inference, we selected the input validation data corresponding to the frequency range for which each model was trained and averaged the results of the four models. For the benchmark 2 inference, we ran all four models on the science data and selected the highest-performing model, as represented in Figure 5. In future work, we aim to develop a single denoising model generalizable to wide frequency ranges.

The hyperparameter of FC Net is listed below:

```
AE(
  (encoder): Sequential(
    (0): Linear(in_features=40000, out_features=4000, bias=True)
    (1): ReLU()
    (2): Linear(in_features=4000, out_features=400, bias=True)
    (3): ReLU()
    (4): Linear(in_features=400, out_features=40, bias=True)
  )
  (decoder): Sequential(
    (0): Linear(in_features=40, out_features=400, bias=True)
    (1): ReLU()
    (2): Linear(in_features=400, out_features=4000, bias=True)
    (3): ReLU()
    (4): Linear(in_features=4000, out_features=40000, bias=True)
  )
)
```

The output of FC Net at every time step is a single float point number. An MSE loss is calculated between the float point number and the ground truth value.

The PU Net model consists of four down layers and four up layers, with contracting paths between each pair of layers. The down layers include Max Pooling and two convolutional operations, while the up layers comprise Deconvolution and Convolution. Additionally, positional encoding is added after each down layer  Vaswani et al. (2017). Lastly, the output is fed into a linear layer to produce

256-dimensional vector at each time step The detailed model hyperparameter could be found within the `network.py` script in `https://anonymous.4open.science/r/TIDMAD`.

The transformer model processes the time series data by using an Embedding layer to encode each input, converting 8-bit integers in the range of $(-128, 127)$ into a 32-dimensional vector. Positional encoding is then added to the embedded time series Vaswani et al. (2017). This augmented data is fed into a Transformer Encoder with two layers, each containing two heads, 128 hidden dimensions, and a $0.1$ dropout rate. Finally, the output is passed through a linear layer to produce a 256-dimensional vector at each time step.

For both PU Net and Transformer, the output at each time step is a 256-dimensional vector, corresponding to 256 possible output classes. This can be considered as a time series semantic segmentation task where there are 256 possible classes to choose from. We adopted Focal Loss in Object Detection to address the class imbalance problem in semantic segmentation task Lin et al. (2017).

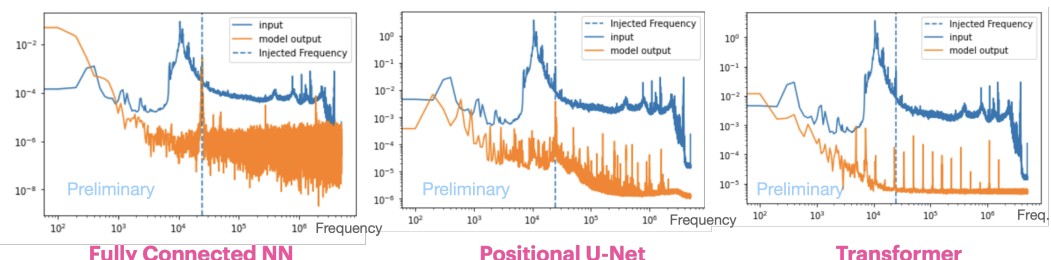

Figure 6: The denoising performance of FC Net (Left), PU Net (Middle), and Transformer (Right) at a single injected frequency. The plot is made by Fourier-transform the time series into frequency space.

The denoising performance of three models at a single injected frequency is illustrated in Figure 6. In this specific instance, the PU-Net model demonstrates superior denoising performance. However, when evaluated using the Denoising Score across all frequencies, the FC Net significantly outperforms the other two models by a large margin.

## C   FREQUENTIST LOG-LIKELIHOOD TEST STATISTICS

The detailed analysis flow to produce the dark matter limit is depicted in Figure 7. The first step involves performing a fast Fourier transform on the time series data in 10-second segments to produce power spectral densities (PSDs). These PSDs are then averaged across the full dataset to generate our average PSD, reflecting the power in the pickup loop as a function of frequency. Since one of the physics parameters, dark matter mass ($m_a$), , is directly proportional to the frequency, the analysis script conducts 11.1 million independent searches for dark matter with varying mass points from 0.4 neV (100 kHz) to 8 neV (2 MHz).

At each mass point, the dark matter limit is obtained using a frequentist log-likelihood ratio test statistic (TS) Foster et al. (2018). Given the local velocity distribution and density of dark matter from astrophysical measurements, as well as our choice of dark matter mass, we create a dark matter signal template for each mass point. These templates are compared to a chunked frequency subset of the average PSD, constructed using a sliding window whose width scales as $\delta f / f \approx 5.5 \times 10^{-6}$. We use equation 1.1 and calibration data to produce the other physics parameter $g_{a\gamma\gamma}$ given detector geometry. By floating the template signal amplitude and allowing the mean background level of noise to vary within each sliding window, we fit the signal template to the data to construct a likelihood as a function of $g_{a\gamma\gamma}$. We then use the TS to determine the $95\%$ one-sided upper limits on $g_{a\gamma\gamma}$ for every mass point Salemi & et. al. (2021); Foster et al. (2018). The resulting limits on $g_{a\gamma\gamma}$ as a function of $m_a$ (black line) as well as the $1/2\sigma$ containment (green, yellow) can be seen on the dark matter sensitivity plot, i.e. "Brazil band" in Figure 7.

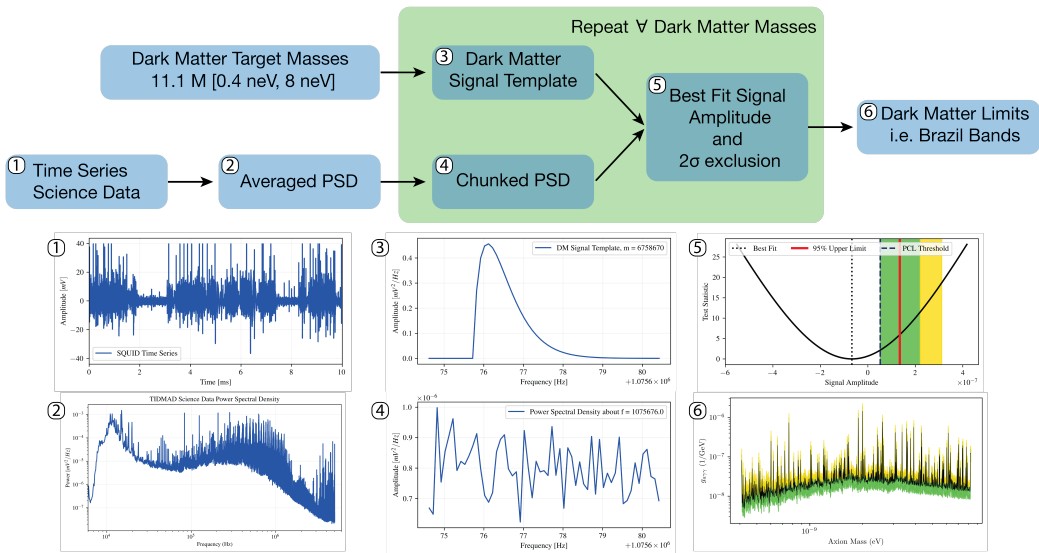

Figure 7: This represents the analysis flow for dark matter science data and the detection logic to build the brazil band limits from time series data (raw or denoised). Key types of data are plotted corresponding to their step in the analysis chain.

## D  CURRENT AXION LIMITS

In Figure 5, the TIDMAD dark matter limits, denoised and raw, are presented alongside ABRA's previous Run 3 limits and various state-of-the-art axion dark matter experiments and observations. In gold, the theoretically motivated couplings for dark matter candidates are highlighted as discussed in Appendix A. In this appendix, a brief summary of the origins of these limits will be provided.

**Light-shining-through-walls:**  In this class of experiments, a high-intensity laser is directed towards a solid barrier. While conventional light cannot traverse the barrier, the interaction with the wall may cause a fraction of the light to convert into axions. As a result of their weak interaction with matter, these axions could pass through the wall. On the opposite side, detectors are placed to identify any reconverted light, which would indicate the presence of axions. Light-shining-through-walls experiments must both create axions from photons and detect these axions by converting them back into photons, whereas ABRA only needs to detect axions, not create them.

**Cavity Haloscopes:**  This class of experiments generate a strong magnetic field to stimulate axions to convert into microwave photons within a resonant cavity-enclosed space. The resonant cavity is finely tuned to amplify specific frequencies of electromagnetic radiation. Sensitive radio receivers then measure the power within the cavity to identify any potential photon signals indicative of axions. While the conversion mechanism is identical to ABRA, the resonant cavity only amplifies targeted frequencies, while ABRA's readout chain has broadband amplification of axion induced signals. ABRA, Cavity Haloscopes, and SHAFT are all examples of Haloscopes – experiments that search for dark matter axions in our galaxy's dark matter halo.

**SHAFT:**  The Search for Halo Axions with Ferromagnetic Toroids (SHAFT) experiment is an axion haloscope with a broadband readout, similar to ABRA in both detection and readout mechanism. There are two main differences between SHAFT and ABRA (1) SHAFT uses toroidal magnets with *ferromagnetic* material in the core to convert the axions (2) SHAFT contains a pairs of stacked ferromagnetic toroids each of which has a separate pickup coil and SQUID readout.

**Astrophysics:**  There are numerous astrophysical processes that would be altered if the axion exists. Broadly, this class of exclusions takes astrophysical observations, calculates how these processes would change if axions exits, and sets limits on possible axion couplings. These limits in-

clude the following astrophysical processes. **Stellar Cooling.** Axions produced in hot astrophysical plasma can transport energy out of stars. This transport of energy critically affects stellar lifetimes, thus observations of stellar energy-loss rates can set limits on axion's couplings to matter. **Photon Flux** Large photon fluxes from astrophysical objects like Supernova 1987A traverse the galaxy before being detected terrestrially. Within the galactic magnetic field, some of these supernova photons could be converted into axions. By observing the gamma-ray signals from such events, strong bounds on axions couplings to photons can be derived. **Black Hole Superradiance.** Light particles, like axions, affect the gravitational waves emitted by black holes through the superradiance mechanism in which axion fields extract energy and angular momentum from the black hole. Observations of stellar black hole spin measurements can therefore constrain allowable axion couplings.

**CAST:** The CERN Axion Solar Telescope (CAST) experiment is a prominent axion helioscope. In contrast to *halo*scopes which search for axions created in the early universe within the dark matter halo surrounding our galaxy, helioscopes search for axions created in our Sun's heliosphere. CAST uses a strong, movable superconducting magnet to convert axions produced in the core of our Sun into into X-ray photons when aligned with the Sun. CAST is equipped with highly sensitive X-ray detectors at both ends of the magnet, designed to capture these photons. By tracking the Sun and searching for excess X-rays that correlate with solar axions, CAST aims to detect solar axions.

# E   DATASHEET

## E.1   MOTIVATION

1. **For what purpose was the dataset creates?** Our datasets were created to train and benchmark ultra-long time series denoising frameworks for the discovery of dark matter.

2. **Who created the dataset and on behalf of which entity?** This dataset is the direct output of the ABRACADABRA detector on behalf of the researchers on the author list.

3. **Who funded the creation of the dataset?** This work was generously funded by the National Science Foundation under grant numbers NSF-PHY-1658693, NSF-PHY-1806440, 2141064.

## E.2   COMPOSITION

1. **What do the instances that comprise the dataset represent?** Each instance represents a voltage at a moment in time read out by our detector. For the SQUID data, this voltage comes from flux on the pickup loop of wire, converted to a voltage by the SQUID detector, read out by our digitizer. For the SG data, this voltage comes directly from a signal generator passed through a power splitter.

2. **How many instances are there in total?** There are 867,260,000,000 voltage instances total.

3. **Does the dataset contain all possible instances or is it a sample of instances from a larger set?** The voltage produced by the SQUID and the SG are continuous. The instances are sampled from this continuous voltage stream at a constant rate of 10MS/s.

4. **What data does each instance consist of?** The data each instance consists of is a raw 8-bit integer from our digitizer. To convert the raw 8-bit integer to a voltage, each bit must be scaled by the ADC voltage i.e. multiply by $40mV/128$.

5. **Is there a label or target associated with each instance?** Yes, for the calibration data, each instance of the SQUID data corresponds to a target which is the instance in the SG data.

6. **Is any information missing form individual instances?** No.

7. **Are relationships between individual instances made explicit?** The instances are related because they come from the same detector just sampled at a different moment in time.

8. **Are there recommended data splits?** Yes, please see Section 2.l

9. **Are there any errors, sources of noise, or redundancies in the dataset?** There are no redundancies. Yes, there are many sources of detector noise.

10. **Is the dataset self-contained, or does it link to or otherwise rely on external resources?** The data is self-contained.

11. **Does the dataset contain data that may be considered confidential?** No

12. **Does the dataset containd ata that, if viewed directly, might be offensive, insulting, threatening, or might otherwise cause anxiety?** No.

Table 2: Summary of critical information about this data release. These are the data files used for training and benchmarking of the baseline algorithms provided.

|  | Training Data | Validation Data | Science Data |
| --- | --- | --- | --- |
| File Name | abra_training _00{00-19}.h5 | abra_validation _00{00-19}.h5 | abra_science _0{000-207}.h5 |
| No. Data Points per File | 2.01e9 | 2.01e9 | 4.01e9 |
| HDF5 File Size | 2.2 GB | 2.2 GB | 2.7 GB |
| ch1 Hardware Input | SQUID | SQUID | SQUID |
| ch2 Hardware Input | SG | SG |  |
| Injected frequencies (Hz) | [1100, 1200, ... , 4.8M, 4.9M] | | |
| Injected amplitudes (mV) | 50 | | |

### E.3 COLLECTION PROCESS

1. **How was the data associated with each instance acquired?** The data associated with each instance is acquired by the ABRACADABRA detector. Full details can be viewed in Sections 1.1 and 2.

2. **What mechanisms or procedures were used to collect the data?** Full details can be viewed in Sections 1.1 and 2 and reference Salemi & et. al. (2021). The hardware necessary for producing said data include, but are not limited to, an Oxford dilution refrigerator, 1T superconducting magnet, two-stage Magnicon SQUID, superconducting pickup loop, superconducting calibration loop, signal generator, digitizer, and data acquisition computer.

3. **If the dataset is sampled from a larger set, what was the sampling strategy?** The voltage produced by the SQUID and the SG are continuous. The instances are sampled from this continuous voltage stream at a constant rate of 10MS/s. The sampling strategy is deterministic and regular.

4. **Who was involved in the data collection process and how were they compensated?** To run the ABRACADABRA experiment, one graduate student was needed. This graduate student was paid via NSF fellowship.

5. **Over what timeframe was the data collected?** The data were collected from 2/21/24 - 2/23/24.

Table 3: Summary of auxiliary files in this data release. These files provide an interesting challenge for the user, however were not used in the training or validation of the baseline models.

|  | Aux Training Data | Aux Validation Data |
| --- | --- | --- |
| File Name | abra_training_00{20-39}.h5 | abra_validation_00{20-39}.h5 |
| No. Data Points per File | 2.01e9 | 2.01e9 |
| HDF5 File Size | 2.2 GB | 2.2 GB |
| ch1 Hardware Input | SQUID | SQUID |
| ch2 Hardware Input | SG | SG |
| Injected Frequencies (Hz) | [1100, 1200, ... , 4.8M, 4.9M] | |
| Injected Amplitudes (mV) | 10 | |

6. **Were any ethical review processes conducted?** No. These data do not involve humans.

7. **Does this dataset relate to people?** No.

### E.4 PREPROCESSING/CLEANING/LABELING

1. **Was any preprocessing/cleaning/labeling of the data done?** No.

### E.5 USES

1. **Has the dataset been used for any tasks already?** No, this dataset has not been used for any tasks yet.

2. **Is there a repository that links to any or all papers or systems that use the dataset?** No. This dataset has yet to be used outside of this paper.

3. **What other tasks could the dataset be used for?** As discussed in Section **??**, these data can generally be used for training time series algorithms. Due to its high coherence and extensive length, it is perfect for cross cutting applications.

4. **Is there anything about the composition of the dataset or the way it was collected and preprocessed that might impact future use?** No.

5. **Are there tasks for which the dataset should not be used?** No.

### E.6 DISTRIBUTION

1. **Will the dataset be distributed to third parties outside of the entity on behalf of which the dataset was created?** Yes, the dataset is open to the public.

2. **How will the dataset be distriuted?** The dataset is publically available to be downloaded from the Open Science Data Federation cache. For download instructions, please see Section F.

3. **When will the dataset be distributed?** The dataset is presently available.

4. **Will the dataset be distributed under a copyright or other intellectual property (IP) license, and/or under applicable terms of use (ToU)?** No.

5. **Have any third parties imposed IP-based or other restriction on the data associated with the instances?** No.

6. **Do any export controls or other regulatory restrictions apply to the dataset or to individual instances?** No.

### E.7 MAINTENANCE

1. **Who will be supporting/hosting/maintaining the dataset?** The dataset is hosted at Open Science Data Federation (OSDF). OSDF also provide distributed cache of the dataset across its global cache location. For more detail, please refer to OSDF Website[1]. The paper authors will be maintaining the dataset.

2. **How can the owner/curator/mangager of the dataset be contacted?** Please email the corresponding author on the author list.

3. **Is there an erratum?** No.

4. **Will the dataset be updated?** No, the dataset will not be updated.

5. **If the dataset relates to people, are there applicable limits on the retention of the data associated with the instances?** This dataset does not relate to people.

6. **Will older versions of the dataset continue to be supported?** Yes, they will continue to be supported.

7. **If others want to extend/augment/build on/contribute to the dataset, is there a mechanism for them to do so?** No. Access to the ABRACADABRA detector is controlled.

---

[1] https://osg-htc.org/services/osdf.html

## F  DATASET AND CODE ACCESS

The data downloading and associated analysis scripts are available at `https://anonymous.4open.science/r/TIDMAD`. All data were uploaded to Open Science Data Federation and cached using their distributed cache system. The TIDMAD dataset can be downloaded using the `download_data.py` script provided in this GitHub repository. This script runs without any external dependencies. This script downloads data by generating a series of wget commands and executing them in a bash environment. `download_data.py` has the following argument:

- `--output_dir -o`: Destination directory where the file will be downloaded, default: current working directory.
- `--cache -c`: Which OSDF cache location should be used to download data. Options include [NY/NorCal/SoCal/Director(default)]:
    - NY: New York
    - NorCal: Sunnyvale
    - SoCal: San Diego
    - Director: automatically find the fastest cache location based on user's location.
        * **WARNING**: Director cache is sometimes unstable. We recommend switching to a different cache if the download fails.
- `--train_files -t`: Number of training files to download, must be an integer between 0 and 20, default 20.
- `--validation_files -v`: Number of validation files to download, must be an integer between 0 and 20, default 20.
- `--science_files -s`: Number of science files to download, must be an integer between 0 and 208, default 208.
- `-f, --force`: Directly proceed to download without showing the file size and asking the confirmation question.
- `-sk, --skip_downloaded`: Skip the file that already exists at `--output_dir`.
- `-w, --weak`: Download the weak signal version of training and validation files. In this version, the injected signal is 1/5 the amplitude of the normal version. This is a more challenging denoising task. Note that the normal version has a file range 0000-0019, while the weak version has a file range of 0020-0039.
- `-p, --print`: Print out all wget commands instead of actually executing the download commands.

In the same github repository, we also provided a `filelist.dat` file which contains line-by-line wget command to download the entire dataset. An example wget command is given here:

```
wget   https://osdf-director.osg-htc.org/ucsd/physics/ABRACADABRA/
ABRA_aires_validation_data/abra_validation_0009.h5
```

## G  CROISSANT METADATA

We created a croissant metadata file `TIDMAD_croissant.json` using protocal presented in Akhtar et al. (2024).

## H  AUTHOR STATEMENT

The authors of this paper all bear responsibility in the case of violation of rights. The information provided in the paper and supplementary material is truthful and accurate. The code from this paper is hosted, managed, and maintained by the paper's first author at `https://anonymous.4open.science/r/TIDMAD`. The data from this paper is hosted, managed, and maintained by the paper authors with download instructions in Section F. The dataset is released under the Creative Commons Attribution (CC BY) license. The code is released under the GNU General Public License (GPL), version 3.

