# OpenReview forum: "TIDMAD: Time Series Dataset for Discovering Dark Matter with AI Denoising"
_ICLR.cc/2025/Conference — ICLR 2025 Conference Withdrawn Submission_

### Official Review · Reviewer_b7fj · 2024-11-03

**Soundness:** 2
**Presentation:** 3
**Contribution:** 2
**Rating:** 3
**Confidence:** 4

**Summary:**

The paper is tackling the problem of creating community standards for experiments searching for dark matter. The goal of this paper is to bring the Physics and ML communities together and help them develop algorithms that would do denoising / segmentation / detection for direct scientific advancements. More specifically, the authors have created a benchmark dataset (consisting of a training, validation, and test/scientific portions). Detecting dark matter has never been done before and would be a huge step forward for humanity. The authors hope that by creating the dataset named TIDMAD and fostering collaboration between researchers in ML and Physics, we will be one step closer to discovering dark matter, which makes up 85% of all mass in the universe. Authors perform experiments using several deep learning architectures to showcase the potential of the dataset and proposed benchmarking scores.
The significance of this work is contributing a new benchmark dataset and benchmarking scores/approaches. However, the paper is not relevant to the ICLR community and would fit better in a physics-oriented venue.

**Strengths:**

The paper is clearly written, technically correct, and seems reproducible given that one has access to the hardware.
The approach is also well-motivated. It is based on the ABRACADABRA experiment, and creates a smaller and possibly more accessible dataset for the community to easily access and run experiments on. The authors create a training set that has ground truth via injecting a sinusoidal signal that spans two orders of magnitude in frequency to represent dark matter signal, and add Gaussian noise via hardware. The authors also create a validation set on the same setup, as well as a test set (they call it the science dataset) that contains no injected ground truth.
The construction of datasets appears to be scientifically rigorous. The proposed benchmarks (1. Denoising score and 2. Dark matter limit) seem to be relevant and rigorous as well. The authors also attempt to show the potential of this dataset by performing denoising using various deep learning architectures, casting a wide net and trying to appeal to a broad audience via using most famous architectures.

**Weaknesses:**

The paper does not present novel findings theoretically or algorithmically, but presents a dataset that could potentially bring impactful findings in the future. Historically, benchmark datasets have been the foundation of advancement in the ML community in the recent decades (e.g.  Imagenet). This work presents a dataset which could potentially have a large impact, however, the paper would be a better fit for audiences other than the ICLR community.
The authors use several architectures and algorithms on the presented dataset. While they use moving average and the SG filter, there are a plethora of classical statistical methods that have not been explored. Before moving to deep learning architectures, authors should do their due-diligence to show how the dataset works with classical statistical methods. While it is a popular thing to use various deep learning architectures, authors should show more explanation reasoning behind this intention -- the approach and justification of applying these architectures and other details about them is lacking.

**Questions:**

1. In my understanding, the science data was created via collecting a signal using the ABRA hardware. In the case when the science data potentially contains dark matter signal, models trained on the training set would hopefully detect the signal in the science dataset. The training data were created via collecting signals in \textbf{the same way}, with the addition of injecting a sine wave. Please elaborate on why this is a good way of collecting training data. Why not simulate the noise as well, and create fully synthetic data? What are the potential advantages and disadvantages of constructing the train data the way you have? If true dark matter signal was picked up by the hardware, then a sine wave was injected, would this method of data construction present any problems (e.g. superposition of waves)?

2. I see that the training set was created via two different amplitudes 50mV (standard) and 10mV (weak), and experiments were presented only on the standard amplitudes but not the weak ones. Were there any experiments done on the weak signals? What were the results? Knowing these results would contribute to the general collective knowledge, even if they were not successful.

3. Authors speak of Gaussian noise, but there is no mention of nose level. It would be helpful to see a concise theoretical setup / data formulation (e.g. noise parameters) where given a certain level Gaussian noise, authors would present the best possible results that optimal methods could achieve. For example, Figure 6 shows that in this setting, PU-Net  performs best. But what does that mean in the general context? What is the \textbf{best possible} outcome, given the noise level present? What about classical statistical methods?

---

### Official Review · Reviewer_Z4LG · 2024-11-03

**Soundness:** 2
**Presentation:** 2
**Contribution:** 2
**Rating:** 3
**Confidence:** 4

**Summary:**

This article describes a data release from the dark matter detector ABRACADABRA. The authors designed two evaluation metrics and experimented five denoising algorithms.

**Strengths:**

Building community datasets and benchmarks in scientific domains is a meaningful effort. The newly launched ABRA experiment's initiative to establish shared resources is encouraging, promoting collaboration, transparency, and progress.

**Weaknesses:**

This paper offers little machine learning novelty. The dataset release is better suited for a dataset track or domain-specific venue, rather than ICLR. Several technical details require further clarification, revision, or investigation (see Questions).

**Questions:**

* How can we ensure that the training and validation data, collected from real experiments, don't contain real signals that could impact the results?
* It is curious that the FC net works better than the transformer. Why did the authors set a significantly smaller model size for the Transformer compared to the FC Net?
* “For both PU Net and Transformer, the output at each time step is a 256-dimensional vector, corresponding to 256 possible output classes.” (Line 765) Why did the authors opt for a different output setting than the FC Net, and what are the implications of this design decision?
* Have you experimented with the hyperparameters of the three neural networks to conclude on their presented architecture?
* Were any validation data used during training and hyperparameter tuning to prevent overfitting?
* “The output of FC Net at every time step is a single float point number.” (Line 751) This does not match the model architecture presented, in which the output dimension is 40,000.

---

### Official Review · Reviewer_RvnM · 2024-11-04

**Soundness:** 3
**Presentation:** 3
**Contribution:** 4
**Rating:** 6
**Confidence:** 4

**Summary:**

The paper introduces TIDMAD, a comprehensive dataset and benchmark derived from the ABRACADABRA experiment aimed at detecting dark matter. The dataset consists of training, validation, and science data subsets and is designed to support machine learning algorithms in denoising ultra-long time series data to improve the search for dark matter. The authors provide benchmark scores to evaluate denoising models and a complete analysis framework for calculating dark matter limits. Various traditional and machine learning-based denoising algorithms were assessed, with the FC Net model achieving the highest performance. The study highlights the potential of machine learning to enhance the sensitivity of dark matter searches, making valuable contributions to both AI and physics communities.

**Strengths:**

- Originality: The creation of a dark matter detection dataset and benchmarking framework is novel and addresses a significant challenge in experimental physics.
- Quality: The paper details the dataset structure, model training, and evaluation metrics with supporting code, demonstrating transparency and reproducibility.
- Significance: The dataset can benefit the broader scientific community by enabling more effective signal extraction in various time series-based applications.
- Clarity: The methodology and experimental setup are well-explained.

**Weaknesses:**

- Baseline Comparisons: The choice of traditional denoising algorithms (e.g., moving average, Savitzky-Golay filter) seems weak, as these performed poorly compared to no processing. Incorporating more sophisticated traditional baselines might strengthen the results.
- Dataset Limitations: The size of the dataset is only 1% of the previous ABRA Run 3 dataset, which may limit the generalizability of conclusions.
- Model Generalizability: The need to train separate models for different frequency ranges indicates that the current deep learning solutions may not generalize well across all data scenarios.

**Questions:**

1. How could more advanced traditional denoising techniques, such as wavelet denoising or Kalman filters, compare with the proposed machine learning models?
2. Is there a strategy for merging the frequency-specific models into a single generalizable model?
3. Can the denoising performance be evaluated using more diverse noise conditions to simulate real-world scenarios more accurately?

---

### Official Review · Reviewer_Wx5G · 2024-11-12

**Soundness:** 2
**Presentation:** 2
**Contribution:** 1
**Rating:** 3
**Confidence:** 4

**Summary:**

This paper presents a dark matter detection dataset and benchmark from a dark matter direct detection experiment. The core problem is about time series denoising. The baseline approaches they present use very basic ML algorithms. The physics experiment is interesting but this paper is clearly a physics paper, written with lots of technical language about the physics experiment, and without any strong argument for where this fits into the landscape of machine learning, or what machine learning people would use this for, and thus is inappropriate for ICLR.

**Strengths:**

- Interesting physics problem that I have not seen explored at all in ML venues.
- The evaluation criteria is interesting and I also don't think this specific metric has been explored much.
- Potentially an interesting problem for a benchmark.

**Weaknesses:**

- This paper is fundamentally a dark matter instrumentation paper that was submitted to an ML venue. It is not appropriate for consideration in this conference. The authors should work with someone in the field of machine learning, or even someone in the physics and machine learning community, so they learn how to frame contributions so that they would be useful for the broader ICLR audience. Even as a physicist myself, I found the paper difficult to understand or to see how to repurpose the dataset for my own work at the intersection.
- It is written heavily with physics terminology with little effort to make it accessible to the audience of the conference.
- Not clear how this work could help advance ML methods.
- Uses basic off-the-shelf machine learning architectures as baselines, without any technical innovations.
- Basic analysis: "we observed that the FC Net model achieved the best performance with a denoising score of 6.43" - without any further general insights presented. No further analysis of why this was true, or what specifically the ML community should pursue in addressing the problem posed by this paper.

**Questions:**

Questions:
- What specific ML research questions would this dataset help enable?

Suggestions:
- I suggest you consider working with someone from the machine learning community to understand how to make such work more impactful for a broader audience.

---

### Note · Authors · 2024-11-14

**Comment:**

We thank the reviewers for their detailed feedback and consideration. The authors recognize that this datasets and benchmark focused paper is not a good fit for ICLR. Thus, we have decided to withdraw our paper.

**Withdrawal Confirmation:**

I have read and agree with the venue's withdrawal policy on behalf of myself and my co-authors.